# Extracellular Vesicles Derived from Young Neural Cultures Attenuate Astrocytic Reactivity In Vitro

**DOI:** 10.3390/ijms23031371

**Published:** 2022-01-25

**Authors:** Daniel Almansa, Héctor Peinado, Raquel García-Rodríguez, Álvaro Casadomé-Perales, Carlos G. Dotti, Francesc X. Guix

**Affiliations:** 1Molecular Neuropathology Unit, Physiological and Pathological Processes Program, Centro de Biología Molecular Severo Ochoa, CSIC/UAM, 28049 Madrid, Spain; daniel.almansa@goumh.umh.es (D.A.); rgarcia@cbm.csic.es (R.G.-R.); acasadome@cbm.csic.es (Á.C.-P.); 2Microenvironment and Metastasis Group, Molecular Oncology Program, Spanish National Cancer Research Centre (CNIO), 28029 Madrid, Spain; hpeinado@cnio.es

**Keywords:** extracellular vesicles (EVs), aging, neural cultures, astrocytes, glial fibrillary acidic protein (GFAP)

## Abstract

Extracellular vesicles (EVs) play an important role in intercellular communication and are involved in both physiological and pathological processes. In the central nervous system (CNS), EVs secreted from different brain cell types exert a sundry of functions, from modulation of astrocytic proliferation and microglial activation to neuronal protection and regeneration. However, the effect of aging on the biological functions of neural EVs is poorly understood. In this work, we studied the biological effects of small EVs (sEVs) isolated from neural cells maintained for 14 or 21 days in vitro (DIV). We found that EVs isolated from 14 DIV cultures reduced the extracellular levels of lactate dehydrogenase (LDH), the expression levels of the astrocytic protein GFAP, and the complexity of astrocyte architecture suggesting a role in lowering the reactivity of astrocytes, while EVs produced by 21 DIV cells did not show any of the above effects. These results in an in vitro model pave the way to evaluate whether similar results occur in vivo and through what mechanisms.

## 1. Introduction

In the last years, many studies unveiled the relevant role that extracellular vesicles (EVs) play in intercellular communication, both in physiological processes and in pathological conditions such as cancer, neurodegeneration, or inflammation [1,2,3]. EVs are small vesicles secreted by every cell type of the body, consisting of a portion of cytoplasm encapsulated by a lipid bilayer. According to a recent classification based on size, EVs are divided into small EVs (sEV), with a size around 100 nm, and large EVs with sizes ranging from 200 nm to 10 μm [4]. Exosomes are a type of EVs belonging to the first group. Exosomes originate from the invagination of the limiting membrane of multivesicular bodies (MVBs), a subtype of late-endosomes [5], towards the lumen of the compartment, which gives rise to intraluminal vesicles (ILVs) that contain cytosolic proteins and nucleic acids. MVBs can fuse to lysosomes for degradation of their content or to the plasma membrane, which results in the release of ILVs to the extracellular space in the form of exosomes.

Numerous studies have demonstrated that sEVs or exosomes can transfer their content to host cells, whether in the immediate vicinity or distantly, by phagocytosis, micropinocytosis, clathrin-mediated endocytosis, caveolin-dependent endocytosis, or membrane–membrane fusion [6]. In the central nervous system (CNS), secreted sEVs are mainly taken up through an endocytosis-based mechanism [7,8] and regulate a sundry of physiological processes in the recipient cell [9,10]. Exosome-mediated PTEN transfer from neurons to glia hampers the proliferation of astrocytes and induces neuronal regeneration [11], and oligodendrocytes-derived exosomes protect neurons from oxidative stress and starvation [12]. Moreover, a large body of evidence points to the anti-inflammatory capacity of sEVs, mainly through the regulation of microglia. In several mouse models of inflammation, sEVs were shown to reduce the activation of microglia and attenuate microglia-induced neuroinflammation [13,14]. Human umbilical MSC-derived sEVs also decreased microglia-mediated neuroinflammation in perinatal brain injury [15], while bone MSC-derived sEVs were shown to be effective in switching the phenotype of microglia from a pro-inflammatory type (M1) to an anti-inflammatory phenotype (M2) on a mouse model of traumatic brain injury (TBI) [16]. Apart from MSC-derived sEVs, other cells of the nervous system secrete sEVs with anti-inflammatory properties. Jiang et al. demonstrated the capacity of neuron-derived exosomes, which carry the microRNA miR-124-3p, to suppress microglial and astroglial activation in a traumatic spinal cord injury (SCI) mouse model [17]. Due to the neuroprotective and anti-inflammatory role played by sEVs, their use as cell-free therapy has been proposed to treat several neurological conditions ranging from mental illness, neurodegeneration, and neuroinflammation [18,19].

In contrast to the large body of evidence that supports the role of sEVs in modulating microglia inflammatory response, there are few reports about the role of sEVs in the regulation of the astrocytic inflammatory activation. Astrocytes become reactive in many pathological conditions of the brain, such as infections, ischemia, or neurodegenerative diseases [20,21,22]. The two main hallmarks of astrocytic reactivity are cell hypertrophy and overexpression of the glial fibrillary acidic protein (GFAP) [23,24]. Reactive astrocytes show a higher number of processes [24] and release pro- (i.e., IL-1β and TNFα) and anti-inflammatory (IL-6 and TGFβ) cytokines, neurotrophic factors, chemokines, complement factors, and reactive oxygen species (ROS) [25,26,27,28]. However, whether or not brain-derived sEVs modulate the reactivity of astrocytes remains less understood. Reactivity of astrocytes is known to increase in aging and under certain conditions [29,30]. Aging couples with increased release of cytokines [29]. Due to the important role of EVs in cell-to-cell communication, sEVs may mediate part of the increased reactivity of astroglia observed in the aging brain. A recent study in our laboratory found an increased secretion of sEVs by neurons aged in culture [31]; however, the effect of these vesicles on other CNS cell types remains unknown.

In this work, we studied the effect of sEVs isolated from neural cells maintained in culture for different lengths of time. We report that sEVs isolated from for 14 days in vitro cells, but not 21 days in vitro, reduce signs of astrocyte reactivity in vitro.

## 2. Results

### 2.1. Characterization of EVs Isolated from 14 and 21 DIV Rat Cortical Neural Culture

Primary neural cultures obtained by chemical and mechanical dissociation of the cerebral cortex of rat embryos contain mostly neurons, but also other cell types such as astrocytes and, to a lesser extent, microglia and oligodendrocytes, which mirrors the multicellular complexity and interactions of the brain (Appendix A). Therefore, although it is still a simplification of the in vivo scenario, this type of culture system represents a suitable model to help us learn more about the biological significance of EVs produced by nerve cells.

Nanoparticle tracking analysis (NTA) and transmission electron microscopy (TEM) analysis of EVs showed that 82% and 85% of EVs isolated from 14 and 21 days in vitro (DIVs) neural cultures, respectively, fall in size ranges from 50 to 150 nm, compatible with the typical size of sEVs such as exosomes [1] (Figure 1A,B). In agreement with our previous findings [31], we observed an increased release of sEVs for 21 DIV cultures in comparison to 14 DIV cultures (Figure 1C), with no significant differences in the size of these vesicles (Figure 1D).

Next, we evaluated the capacity of neurons and astrocytes in vitro to take up exogenous sEVs added to the media of neural cultures. For this purpose, sEVs were isolated from the media of 14 DIV neural cultures and labeled with the hydrophobic probe Bodipy-cholesterol (Figure 2A), which binds to the lipid bilayer of the vesicles. After verifying by TEM that structure and number were preserved during this procedure (Figure 2B) and that they were labeled correctly (Figure 2C), isolated vesicles were added to the media of neural cultures. Confocal microscopy analysis revealed that both neurons and astrocytes internalized exogenous bodipy-labeled sEVs, which were preferentially concentrated in the cell body (Figure 2D).

### 2.2. EVs Isolated from 14 or 21 DIV Rat Cortical Neural Cultures Produce Differential Deleterious/Beneficial Consequences When Added to 14 DIV Cultures

To gain insight into the biological role of sEVs secreted by neural cultures, 14 DIV neural cultures were treated for 24 h with sEVs obtained from the media of 14 DIV or 21 DIV neural cultures. In order to increase the chances of obtaining a biological effect, the sEVs were added at 3 times the concentration present in the media of 14 DIV cultures (mean ± 95% Confidence Interval = 24.7 × 10^3^ ± 15 × 10^3^ number EVs/cell in physiological conditions versus 82.8 × 10^3^ ± 58 × 10^3^ number EVs/cell in treatment conditions). We first evaluated the effect of sEVs on the levels of tau protein and the secretion of Aβ peptide, two proteins tightly related to neurodegeneration [32,33,34]. EVs are known to spread phosphorylated forms of tau protein throughout the brain of Alzheimer’s disease (AD) patients [32,33] and contain Aβ peptide, responsible for some of the signs of cell toxicity of AD [34]. Western blot analysis of cell lysates revealed no changes in the levels of total tau or PHF-tau (a phosphorylated form of tau associated with AD) in neural cultures treated with 14 or 21 DIV sEVs (Figure 3A,B). In addition, no changes in the secretion of Aβ40 and Aβ42 peptides, determined by ELISA, were found in 14 DIV neural cultures untreated or treated with 14 or 21 DIV EVs (Figure 3C). Next, we evaluated the effect of exogenous EVs on the levels of the synapse-associated postsynaptic density protein PSD95 (Figure 3D) and the presynaptic protein synaptophysin (Figure 3E). Again, no differences were found after the treatment with 14 or 21 DIV EVs (Figure 3D,E). The same negative results were obtained for the phosphorylated form of p38 MAPK, a cellular stress maker (Figure 3F), or for the TrkB-Akt survival pathway (Figure 3G,H). Notwithstanding, 14 DIV neural cultures treated with 14 DIV EVs showed a significant decrease in the release of lactate dehydrogenase (LDH) to the media (Figure 3I), indicative of a decrease in cellular stress. This effect was not observed in 14 DIV cultures treated with EVs from 21 DIV cultures.

To determine if the absence of negative effects was not the consequence of too short a treatment, we repeated the same analysis by exposing 14 DIV cells to sEVs for 48 h. Again, no significant differences were observed (Figure 4A–H), except for a significant reduction of LDH in cultures treated with 14 DIV EVs, which was not observed with the 21 DIV EVs treatment (Figure 4I).

Next, we proceeded to study if the treatment with EVs induced any morphological alterations in neurons. Morphometric analysis revealed that EVs from 14 DIV cultures, but not from 21 DIV cells, increased the number of secondary branches, while 21 DIV EVs reduced the number of primary and secondary branches in neurons (Figure 5), which, as a whole, would indicate that the content and/or the mechanism of action of these vesicles change dramatically over time in culture, going from being beneficial to adverse.

### 2.3. sEVs from 14 DIV Neural Cultures Affect Morphology and Reactivity of Astrocytes

The morphological changes in neurons and the presence of a lower LDH release motivated us to determine in a second stage the effect of sEVs on astrocytes and microglia, the main mediators of the brain’s inflammatory response. As in the previous experiments, treatments with EVs were performed using 3 times the basal concentrations found in 14 DIV neural cultures. While non-significant differences were found for the expression levels of the microglial marker IBA1 (Figure 6A,B), we found a significant reduction of the glial fibrillary acidic protein (GFAP), an astrocyte-specific marker [23]. Importantly, this effect was observed with EVs from 14 DIV cells but not with EVs isolated from 21 DIV cultures (Figure 6A,C). Importantly, the effect on the GFAP expression levels was not due to changes in the content of GFAP in the EVs isolated from the different cultures (Appendix A).

Since GFAP levels can alter the morphology of astrocytes [35,36], we next compared the arborization of astrocytes after treatment with EVs (Figure 7A). Morphometric analysis revealed that EVs from 14 DIV cultures, but not from 21 DIV cells, reduced the number of secondary branches, while no differences were found for the number or the length of primary processes (Figure 7A–D) and for the density of primary branches (Figure 7E). The reduction in the number of secondary branches, together with the decrease of the GFAP expression levels, suggests a role for EVs in decreasing astrocytic reactivity. To test this, we compared the effect of EVs on the release of a series of cytokines by 14 DIV neural cultures. While EVs from both 14 and 21 DIV cultures reduced slightly the secretion of the cytokines IFNγ and IL-1a (Appendix A), only EVs from 14 DIV cultures decreased the levels of IL-6 (Appendix A), a cytokine predominantly released in neurodegenerative diseases through EVs [37] in the CNS and involved in astrogliosis. Although the changes observed did not reach statistical significance, altogether, our data support the role of EVs released by young neural cultures in reducing astrocytic reactivity. 

## 3. Discussion

In this work, we used primary cortical neural cultures to study the biological consequences of sEVs derived from young and aged cells. Although we only studied a limited series of biological scenarios, one of them was clearly affected after adding sEVs to the cells: reduced astrocyte reactivity. However, this effect was clearly dependent on the age of the sEVs producing cells. Thus, sEVs produced by 14 DIV cells but not 21 days reduced the expression of GFAP and decreased the astrocytic arbor complexity, two hallmarks of the reactivity of astrocytes [23,24]. This is in agreement with recent studies suggesting a role for EVs reducing the reactivity of astrocytes induced by LPS [38] and in a traumatic spinal cord injury (SCI) mouse model [17]. The observation that this type of seemingly beneficial effect does not occur when cultures are incubated with vesicles produced by 21 DIV cultures implies that cells aging in vitro stop producing beneficial compounds, in addition to producing toxic compounds.

The reduction of GFAP levels has proved effective in preventing neurons from dying. Menet et al. showed that neurons co-cultured with GFAP-deficient astrocytes had better survival and increased neurite outgrowth in comparison to neurons co-cultured with WT astrocytes [39,40], and vimentin, another protein constitutive of the astrocytic intermediate filaments, was not required for this effect. Another study carried out by Hanbury et al. showed that GFAP knockout astrocytes protect striatal neurons from excitotoxic or metabolic insult by secreting the glial cell-derived neurotrophic factor [41]. These findings are in agreement with our results showing decreased cell death and higher neuronal ramification in neural cultures treated with 14 DIV EVs. In addition, in agreement with our observations, GFAP expression is required for the formation of astrocytic networks in other systems [35], and astrocytes deficient for GFAP show reduced arbor complexity with a reduced number of processes [42]. It would be interesting to compare the content of Aldolase C in EVs from 14 and 21 DIV cultures since it was demonstrated that Aldolase C-containing EVs derived from astrocytes reduce the dendritic complexity of neurons [43].

Our study also provided some clues to the possible mechanism through which sEVs influence astrocyte behavior. We showed that the media of 14 DIV EVs- or 21 DIV EVs-treated neural cultures contain different cytokines. Thus, IL-1β and MIP decrease only after treatment with 21 DIV EVs, while INFγ, IL-1α, and MCP-1 are reduced by the treatment with both 14 DIV and 21 DIV EVs. Remarkably, the levels of IL-6 were only reduced by 14 DIV EV. The differential effects on the expression of different cytokines by 14 and 21 DIV EVs may indicate that sEVs from young and old brains modulate inflammation in distinct ways, though further investigation is required.

Importantly, our results in vitro on IL-6 deserve special attention. Previous works have shown that IL-6 levels mostly depend on astrocyte reactivity [44]. Similarly, IL-6 can induce either neuroprotection or neuroinflammation, depending on the context. Thus, it was shown that IL-6 is required for the differentiation of glial cells, participates in the regeneration of peripheral nerves, and acts as a neurotrophic factor [45,46,47,48]. In the classical signaling pathway, IL-6 interacts with the IL-6 receptor (IL-6R) on the surface of the target cell, which induces the dimerization of the cytokine receptor gp130, triggering anti-inflammatory signals [49]. However, many cell types, such as astrocytes, oligodendrocytes, and most neuronal types, lack IL-6R on the surface [50,51,52]. These cells can be activated by the trans-signaling pathway through the formation of a complex between a soluble version of the IL-6R receptor and IL-6 that can directly induce the dimerization of gp130. Contrary to classical signaling, the trans-signaling pathway is pro-inflammatory and responsible for IL-6-induced neurodegeneration [53]. Indeed, an increase in the IL-6 levels is seen in several neurological conditions, such as Alzheimer’s disease (AD), Parkinson’s disease, and Multiple Sclerosis [54,55,56], and elevated plasma levels of IL-6 increase the risk of developing dementia [57] and precedes the onset of AD [58].

Aside from the modulatory effect on inflammation, EVs from 14 DIV or 21 DIV cultures could contain different levels of specific biomolecules that could affect the level of expression of GFAP, such as neuregulin-1 (NRG1) or ligands of notch (both of them have been identified previously in EVs) [59,60,61]. More research is needed to determine the true nature of the factor(s) responsible for 14-day culture-derived EVs exerting the beneficial effect on astrocytes. We also need more research to determine if the lack of a beneficial effect from 21 DIV EVs is due to the presence of deleterious factors or simply because the beneficial factors are produced to a lesser extent or are less active. Our results in Figure 5 show that 21 DIV EVs decrease neuronal branching, which suggests a potential gain-of-toxic-function of EVs with age.

In summary, we showed that sEVs isolated from young cell neural cultures increase neuronal branching and reduce signs of astrocyte reactivity in vitro as opposed to sEVs isolated from aged cell cultures. If the results presented here were to occur in situ, we would be facing a scenario in which a part of the physiological increase in brain inflammation with age would be a consequence of the gradual loss of the natural anti-inflammatory capacity present in EVs generated at young ages. Future work will determine whether this scenario is real.

## 4. Materials and Methods

### 4.1. Rat Primary Cortical Cultures

Cortical neurons obtained from rat embryos [62] present numerous morphological and functional characteristics of cortical neurons in situ. Primary cultures of brain cortex were prepared from embryonic day 18 (E18) Wistar rats as described in Dichter, 1978 [54]. Cortex was dissected and placed into ice-cold Hanks solution (Hanks Buffer Salt Solution Ca^2+^ and Mg^2+^ free, Thermo Fisher Scientific, Waltham, MA, USA) supplemented with 7 mM HEPES and 0.45% glucose. The tissue was then treated with 0.005% trypsin (trypsin 0.05% EDTA; Gibco, Thermo Fisher Scientific, Waltham, MA, USA) and DNase (72 mg/mL; Merck MilliporeSigma, Burlington, MA, USA) incubated at 37 °C for 16 min. Cortex was washed three times with Hanks solution. Cells were dissociated in 5 mL of plating medium (Minimum Essential Medium supplemented with 10% horse serum and 20% glucose), and cells were counted in a Neubauer Chamber. Cells were plated into dishes pre-coated with 0.1 mg/mL or 0.5 mg/mL poly-D-lysine (Merck MilliporeSigma), for biochemistry or immunofluorescence experiments, respectively, 300,000 cells/well in a 6 multi-well plate for biochemistry and 100,000 cells/well in a 6 multi-well plate for immunofluorescence, and afterwards, they were placed into a humidified incubator containing 95% air and 5% CO_2_. The plating medium was replaced with equilibrated neurobasal media supplemented with B27 and GlutaMAX (Gibco, Thermo Fisher Scientific) after 4 h.

On DIV 7, the culture medium was progressively replaced (1/4 of the media each day for 4 days, the fourth day being almost totally replaced) with MEM media supplemented with N2 (Gibco, Thermo Fisher Scientific) and without GlutaMAX. Cortical neural cultures were kept the days corresponding to each condition.

### 4.2. Isolation of Extracellular Vesicles

With the aim of evaluating the biological effects of adding sEVs on neurons and astrocytes maintained in vitro, sEVs were isolated by differential centrifugation from the media of cortical neural cultures for different lengths of time (Appendix A). The 14 or 21 DIV neural culture conditioned media (25 mL of MEM media with N2 supplement), collected from one 150 mm culture dish (4.8 × 10^6^ cells), was subsequently centrifuged at 200× *g* and 2000× *g* (10 min each) at 4 °C in order to remove dead cells and debris. After transferring the supernatant to a fresh tube (Cat #344058, Beckman Coulter, Brea, CA, USA), the final volume was made up to 38 mL with filtered PBS 1×. Next, the supernatant was centrifuged at 10,000× *g* and 4 °C for 30 min in a TST28.38 rotor (Kontron AG, Augsburg, Germany) to remove larger microvesicles. A total of 34 mL of the supernatant was transferred to a tube (Cat #344058, Beckman Coulter) on the top of 4 mL of a 30% sucrose/PBS layer [63] and centrifuged at 100,000× *g* and 4 °C for 2 h in a TST28.38 rotor (Kontron AG, Augsburg, Germany). The supernatant was removed, leaving 6 mL volume at the bottom of the tube, corresponding to the 4 mL sucrose plus 2 mL of media, which was transferred to a new tube (Cat #344058, Beckman Coulter). PBS was added up to 38 mL, and samples were centrifuged at 100,000× *g* and 4 °C for 90 min in a TLA100.1 rotor (Beckman Coulter). Finally, the pellet containing EVs was re-suspended in filtered PBS 1× for EM analysis or treatment of cultures; otherwise, RIPA lysis buffer (see formulation in Western Blot Analysis section) with 1× protease and phosphatase inhibitors was used to lysate the EVs for Western blot analysis (Appendix A).

### 4.3. Nanoparticle Tracking Analysis

A total of 1 mL of the conditioned media (MEM supplemented with N2 supplement) of rat primary cortical neural cultures (seeded in 150 mm culture dishes at a density of 4.8 × 10^6^ cells/dish and 25 mL of media) was subsequently centrifuged at 200× *g* and 2000× *g* for 10 min each in order to remove dead cells and debris. The supernatant was transferred to a new Eppendorf tube and centrifuged at 10,000× *g* and 4 °C for 30 min. The supernatant was diluted 1:10 in filtered PBS 1× to obtain a concentration in the range of 10^8^ and 10^9^ vesicles/mL, and the concentration of vesicles present in the sample was analyzed by Nanoparticle Tracking Analysis (NTA) with a NanoSigh NS500 instrument (Malvern Panalytical, Malvern, UK). The instrument was equipped with a 488 nm laser, a high sensitivity CMOS camera. Both the concentration of the vesicles present in the media of neural cultures at different DIV, as well as the mean size, were analyzed using the nta2.3 software (Malvern Panalytical) after filming 3 × 60 s long videos.

### 4.4. Bodipy Labeling of Isolated EVs Using BODIPY-Cholesterol™

The EVs isolated from 14 and 21 DIV neural cultures were labeled using a lipophilic dye, Bodipy-cholesterol (TopFluor^®^ Cholesterol 23-(dipyrrometheneboron difluoride)-24-norcholesterol; #810255, Avanti Polar Lipids, Inc., Birmingham, AL, USA). The EVs were incubated with 1 µM Bodipy-cholesterol in PBS for 1 h at 37 °C in 5% CO_2_. Unincorporated dye was removed using Invitrogen™ spin columns (MW 3000, Gibco, Thermo Fisher Scientific) according to the manufacturer’s instructions. Subsequently, the labeled EVs were immediately added to the cell cultures for exosomes uptake analysis. As a negative control, unlabeled EVs underwent the same procedure. Bodipy-cholesterol alone was also column-purified, and the fluorescent signal of the eluate (after removal of unbound dye) was subtracted from the fluorescent signal of unlabeled or Bodipy-cholesterol-labeled EVs.

### 4.5. Cell Treatments

For cell treatment, 14 DIV rat primary cortical neural cultures were seeded on 6-wells plates (300,000 cells per well for biochemistry analysis and 100,000 cells per well for the immunocitofluorescence study) pre-coated with poly-D-lysine (Merck MilliporeSigma; 0.5 mg/mL poly-D-lysine in borate buffer for immunocitofluorescence experiments; 0.1 mg/mL poly-D-lysine in borate buffer for biochemistry analysis). Cultures were treated for 24 h or 48 h with EVs isolated from 14 or 21 DIV rat primary cortical neural cultures. Before each treatment, half of the media (1 mL) was replaced by fresh MEM media supplemented with N2 (Gibco; Thermo Fisher Scientific); the final volume of the media was 2 mL/well. The same procedure was applied to control wells except for the EVs treatments.

### 4.6. Immunofluorescence Analysis

Cells were washed with PBS 1× at room temperature (RT) and fixed in 4% paraformaldheide (PFA) diluted in PBS 1× for 10 min at RT and in the dark. Then, PFA was replaced by PBS 1×, and samples were kept at 4 °C until processing for immunofluorescence analysis. On the day of the experiment, cells were permeabilized for 10 min with 0.1% Triton X-100 in PBS 1×, then blocked for 15 min with 3% Bovine Serum Albumin (BSA) in PBS 1×. Coverslips were incubated with the primary antibodies diluted in PBS 1× for 2 h and at RT: anti-GFAP 1:500 (MAB3402, Merck MilliporeSigma); anti-MAP2 1:500 (822501, Biolegend, San Diego, CA, USA); anti-Oligo2 1:500 (R&D Systems, Minneapolis, MN, USA). Next, coverslips were washed 3 times with PBS 1× and incubated for 1h with an Alexa555-conjugated anti-mouse antibody (1:1000; Thermo Fisher Scientific) and Alexa488-conjugated anti-chicken antibody (1:1000; Thermo Fisher Scientific). After a 5 min DAPI staining (1 µg/mL in PBS 1×, Merck MilliporeSigma) and 3 washes with PBS 1×, coverslips were finally fixed using Mowiol-DABCO gel mounting agent (Merck MilliporeSigma). 

The preparations were analyzed with an LSM 710 confocal microscope system (Zeiss, Oberkochen, Germany). Zeiss imaging software was used to analyze the confocal images. 

### 4.7. Cell Culture Lysates

At 24 h or 48 h after treatment with EVs, the cell media was removed, and the cells were washed once with PBS 1× and lysed in ice-cold RIPA buffer (20 mM Tris-HCl, pH 7.5, 150 mM NaCl, 1 mM EDTA, 1 mM EGTA, 1% NP-40, 1% sodium deoxycholate, 0.1% SDS) containing 1× phosphatase inhibitors (Merck MilliporeSigma) and 1× proteases inhibitors (cOmpleteTM, Merck MilliporeSigma).

### 4.8. Cytokines Quantification

At 24 h after treatment with EVs, the cell media was used to quantify the cytokine levels with the help of a commercial Rat inflammation ELISA Strip (EA-1201, Signosis, Santa Clara, CA, USA), as described in the manufacturer’s protocol in order to get the profile of 8 cytokines (TNFα; IL-6, IFNγ, IL-1a, IL-1b, MCP-1, Rantes, and MIP). Absorbance at 450 nm was read in a FLUOstar OPTIMA microplate reader (BMG LABTECH, Ortenberg, Germany)

### 4.9. LDH Assay

At 24 h or 48 h after treatment with EVs, 50 µL of cell media per sample was used to determine the lactate dehydrogenase (LDH) levels with the Pierce LDH Cytotoxicity Assay Kit (88954, Thermo Scientific) following the manufacturer’s instructions, in order to quantify cell death. Conversion of LDH to formazan is detected by reading the absorbance at 490 nm in a FLUOstar OPTIMA microplate reader (BMG LABTECH).

### 4.10. Aβ Determination

For detection of rat Aβ42, the media was probed with a Rat beta-Amyloid (42) ELISA Kit (#290-62601, Fujifilm Wako Chemicals, Osaka, Japan), as described in the manufacturer’s protocol. A standard curve with the following Abeta42 concentrations was built in order to determine the Abeta42 levels in the samples: 100 pM, 50 pM, 25 pM, 10 pM, 5 pM, 2.5 pM, 1 pM, and 0 pM. Oxidized TMB substrate is detected by reading the absorbance at 450 nm in a FLUOstar OPTIMA microplate reader (BMG LABTECH).

### 4.11. Western Blots Analysis

Cells or EVs were lysed as described above (Section 4.8 of Materials and Methods). Proteins were prepared in Laemmli buffer (Tris-HCl 25 mM pH 6.8, sodium dodecyl sulfate (SDS) 1%, glycerol 3.5%, 2-mercaptoethanol 0.4%, and bromphenol blue 0.04%) and separated by electrophoresis in polyacrylamide gels in the presence of SDS at a constant voltage. Subsequently, they were transferred onto nitrocellulose membranes, and after blocking with blocking solution (1% bovine serum albumin (BSA) in 0.1% Tween-20 in TBS (T-TBS)), membranes were incubated with the corresponding primary antibody (see Table 1 below) diluted in blocking buffer overnight at 4 °C. After washing the membranes with T-TBS, they were incubated with the relevant secondary antibodies coupled to horseradish peroxidase and diluted 1/2,500 for 1 hr at RT. The proteins recognized by the antibodies were detected with luminol (Pierce ™ ECL Western Blotting Substrate, Thermo Fisher Scientific), and chemiluminescence was measured using a CCD camera (Amersham Imager 680, GE Healthcara, Chicago, Illinois, USA). The bands corresponding to the proteins of interest were densitometrated by the FIJI digital image processing software and were normalized with respect to the values obtained for the charge control protein actin.

### 4.12. Determination of Protein Concentration

The concentration of proteins present in the homogenates was determined by means of the BCA assay (Pierce™ BCA Protein Assay kit, Thermo Fisher Scientific), following the indications of the commercial kit. Samples were diluted at 1:5 in Milli-Q water. The following Bovine Serum Albumine (BSA) standard curve was build to determine the protein concentration of samples: 2 mg/mL, 1.5 mg/mL, 1 mg/mL, 0.75 mg/mL, 0.5 mg/mL, 0,25 mg/mL, 0.125 mg/mL, 0.025 mg/mL, 0 mg/mL. Protein concentration was determined by reading the absorbance at 560 nm in a FLUOstar OPTIMA microplate reader (BMG LABTECH). 

### 4.13. Transmission Electron Microscopy (TEM)

Isolated EVs were placed on glow discharged 200-mesh formvar/carbon copper grids and stained with uranyl acetate 2% in water for 1.5 min at RT, followed by 3 washes in H_2_O. The microscope used to analyze the samples was Jeol Jem-1010 (Jeol, Tokyo, Japan), and pictures were taken with the camera 4K × 4K F416 from TVIPS (TVIPS, Gauting, Germany).

## Figures and Tables

**Figure 1 ijms-23-01371-f001:**
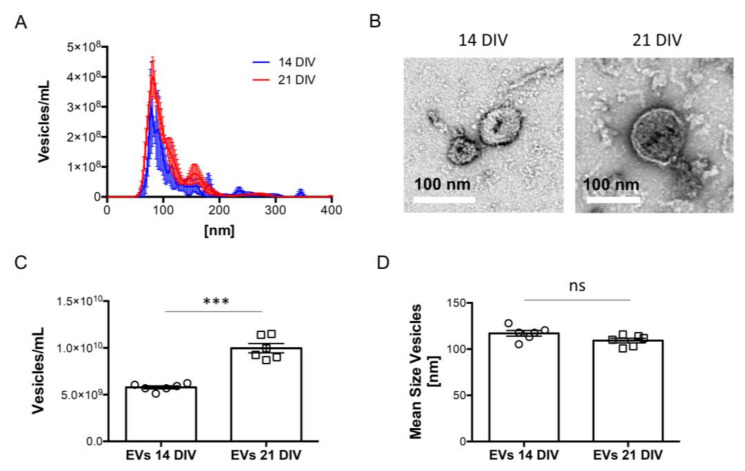
Characterization of extracellular vesicles (EVs) isolated from 14 and 21 DIV rat cortical neural cultures. EVs samples from neuronal culture media were processed by serial centrifugation, as described in “Materials and Methods”. (**A**) Plot showing the size distribution of EVs secreted by 14 and 21 DIV rat cortical neural cultures and quantified in the post-10,000 g supernatant through Nanoparticle tracking analysis with the help of a NanoSight NS500 instrument. (**B**) Representative Transmission Electron Microscopy (TEM) images of EVs isolated from the media of 14 and 21 DIV neural cultures using the protocol shown in “Materials and Methods”. Scale bars on both images represent 100 nm. (**C**) Plot comparing the total number of EVs present in the media of 14 and 21 DIV neural cultures, determined from the area under the curve (AUC) of size distribution profiles like the ones shown in panel A. Data are expressed as vesicles/mL, and the bars represent mean ± SEM. Statistical significance was analyzed by two-tailed unpaired *t*-test (*** *p* < 0.001, n = 6). (**D**) Plot comparing the size of EVs present in the media of 14 and 21 DIV neural cultures, determined from size distribution profiles as the ones shown in panel A. Data are expressed as the mean size of vesicles, and the bars represent mean ± SEM. Statistical significance was analyzed by a two-tailed unpaired *t*-test (ns = non-significant, n = 6).

**Figure 2 ijms-23-01371-f002:**
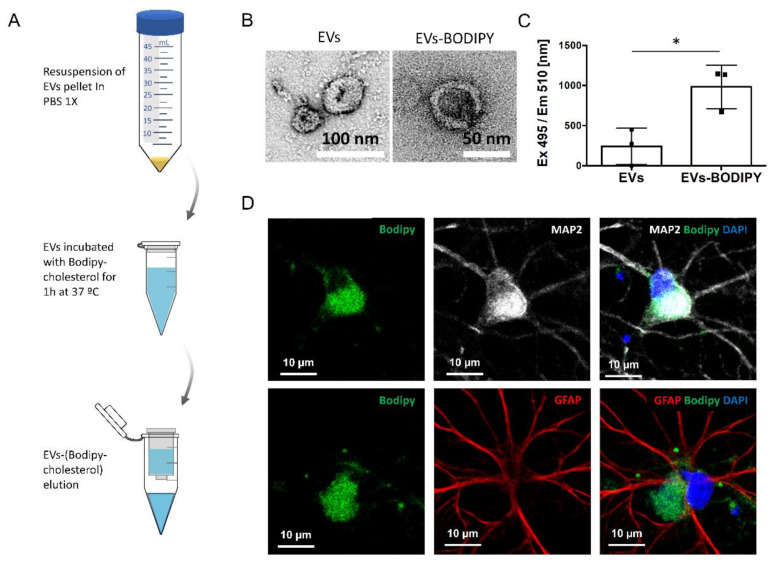
Confocal analysis of Bodipy-cholesterol-labeled EVs uptake by neurons and astrocytes. (**A**) Schematics of the procedure followed to label EVs using Bodipy-cholesterol. EVs in PBS are incubated with 1 µM Bodipy-cholesterol diluted for 1 h at 37 °C. Subsequently, the unincorporated Bodipy-cholesterol was removed using Invitrogen™ spin columns (MW 3000, Thermo Fisher). Bodipy-cholesterol-labeled EVs were recovered in the eluted fraction. (**B**) Representative Transmission Electron Microscopy (TEM) images of unlabeled EVs or Bodipy-cholesterol labeled EVs derived from the media of 14 DIV neural cultures after elusion from the Invitrogen™ spin columns. (**C**) Plot comparing the fluorescence (Excitation 495 nm/Emission 510 nm) of unlabeled EVs (autofluorescence) or Bodipy-cholesterol-labeled EVs using the protocol schematized in panel A. The graph shows the mean fluorescence ± SEM. Statistical significance was analyzed by a two-tailed unpaired *t*-test (* *p* < 0.05, n = 3). (**D**) Representative confocal images of 14 DIV neural cultures treated for 24 h with Bodipy-cholesterol-labeled EVs isolated from the media of 14 DIV neural cultures. Both neurons (stained with the neuronal marker MAP2, in white) and astrocytes (stained with the astrocytic marker GFAP, in red) show the capacity to take up Bodipy-cholesterol-labeled EVs (Bodipy, in green). Nuclei are stained with DAPI. Scale bars represent 10 µm in all the images.

**Figure 3 ijms-23-01371-f003:**
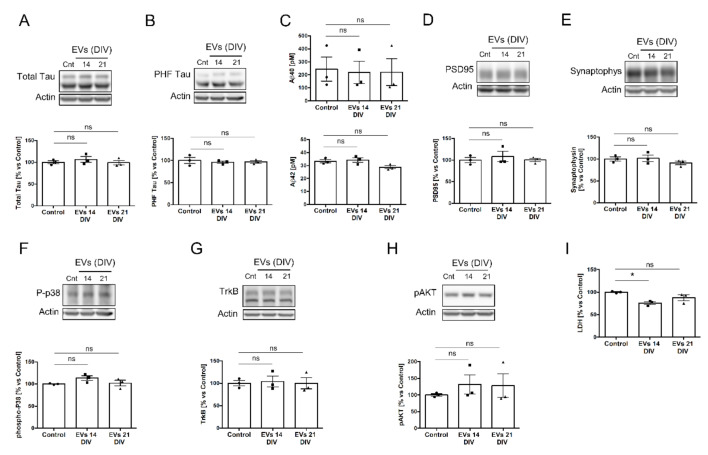
Biological effect of 14 and 21 DIV isolated EVs on 14 DIV rat cortical neural culture for a 24 h treatment period. (**A**) Western blot analysis of total tau protein in 14 DIV neural culture lysates untreated (cnt) or treated for 24 h with EVs isolated from the media of 14 DIV or 21 DIV neural cultures. Actin was used as loading control. The plot shows the levels for total tau protein normalized by actin and quantified from Western blot experiments as the ones shown in this panel. Graph shows the mean protein levels relative to the untreated (control) condition ± SEM (**B**) Same analysis as in panel A, but for tau phosphorylated at S396/T404 (PHF-1 epitope). (**C**) Plots comparing the Aβ40 levels (upper graph) or the Aβ42 levels (lower graph) in the media of rat cortical neural cultures untreated (control) or treated for 24 h with EVs isolated from the media of 14 DIV or 21 DIV neural cultures. Graph shows the Aβ concentration (pM) ± SEM. (**D**) Same analysis as in panel A was performed for the postsynaptic protein PSD95, (**E**) the presynaptic protein synaptophysin, (**F**) the phosphorylated form of p38, (**G**) the receptor for neurotrophins TrkB, and (**H**) the phosphorylated form of AKT at S473. (**I**) The plot compares the relative levels of lactate dehydrogenase (LDH) measured in the media of 14 DIV neural cultures untreated (control) or treated for 24 h with EVs isolated from the media of 14 DIV or 21 DIV neural cultures. All graphs show the mean LDH levels relative to the control condition ± SEM. Statistical significance was analyzed by one-way ANOVA. Post hoc analysis was analyzed by Tukey’s Multiple Comparison Test (* *p* < 0.05; ns = non-significant, n = 3 independent experiments). Note: the bands for actin are the same for panels A and H since the same Western blot was used to analyze the expression levels of total tau and p-AKT.

**Figure 4 ijms-23-01371-f004:**
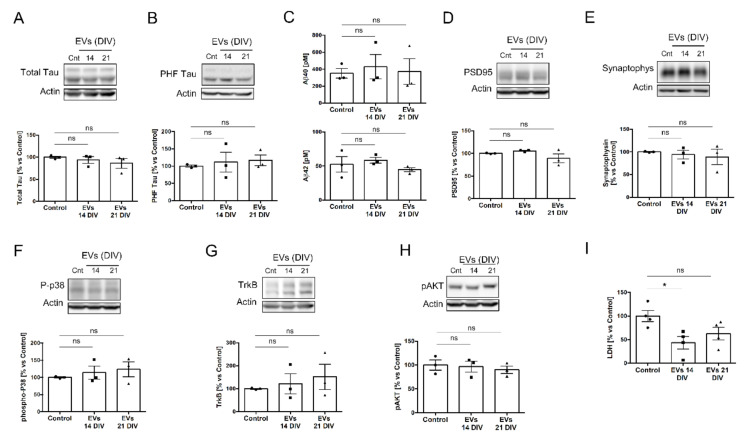
Biological effect of 14 and 21 DIV isolated EVs on 14 DIV rat cortical neural culture for a 48 h treatment period. (**A**) Western blot analysis of total tau protein in 14 DIV neural culture lysates untreated (cnt) or treated for 48 h with EVs isolated from the media of 14 DIV or 21 DIV neural cultures. Actin was used as loading control. The plot shows the levels for total tau protein normalized by actin and quantified from Western blot experiments as the ones shown in this panel. Graph shows the mean protein levels relative to the untreated (control) condition ± SEM (**B**) Same analysis as in panel A, but for tau phosphorylated at S396/T404 (PHF-1 epitope). (**C**) Plots comparing the Aβ40 levels (upper graph) or the Aβ42 levels (lower graph) in the media of rat cortical neural cultures untreated (control) or treated for 48 h with EVs isolated from the media of 14 DIV or 21 DIV neural cultures. Graph shows the Aβ concentration (pM) ± SEM. (**D**) Same analysis as in panel A was performed for the postsynaptic protein PSD95, (**E**) the presynaptic protein synaptophysin, (**F**) the phosphorylated form of p38, (**G**) the receptor for neurotrophins TrkB, and (**H**) the phosphorylated form of AKT at S473. (**I**) The plot compares the relative levels of lactate dehydrogenase (LDH) measured in the media of 14 DIV neural cultures untreated (control) or treated for 48 h with EVs isolated from the media of 14 DIV or 21 DIV neural cultures. All graphs show the mean LDH levels relative to the control condition ± SEM. Statistical significance was analyzed by one-way ANOVA. Post hoc analysis was analyzed by Tukey’s Multiple Comparison Test (* *p* < 0.05; ns = non-significant, n = 3 independent experiments). Note: the bands for actin are the same for panels D and F since the same Western blot was used to analyze the expression levels of total PSD95 and p-p38.

**Figure 5 ijms-23-01371-f005:**
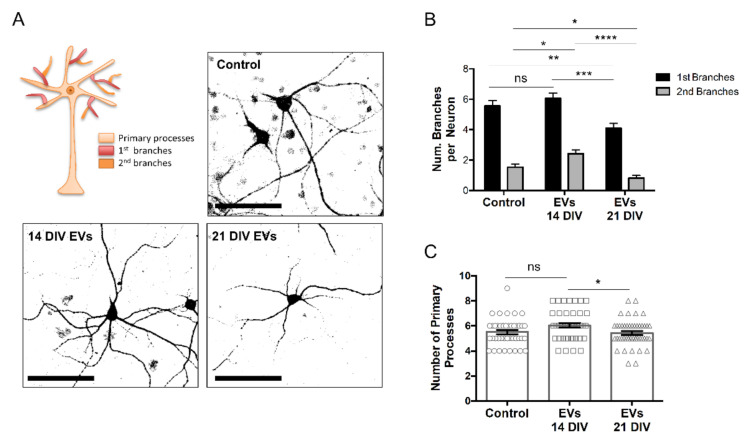
Effect of 14 and 21 DIV isolated EVs on neuronal branching. (**A**) Representative confocal images of neuronal processes, detected with an antibody against the dendritic marker MAP2, of 14 DIV neural culture untreated (control) or treated for 24 h with EVs isolated from the media of 14 DIV (EVs 14 DIV) or 21 DIV (EVs 21 DIV) neural cultures. A scheme highlighting the primary processes, primary branches, and secondary branches of neurons is shown on the left upper quadrant of this panel. 14 DIV neural cultures untreated (control) or treated for 24 h with EVs 14 DIV or EVs 21 DIV were analyzed for the number of primary and secondary branches per neuron (bar = 100 µm) (**B**,**C**) number of primary processes. All graphs show mean values ± SEM. Statistical significance was analyzed by one-way ANOVA. Tuckey’s Multiple Comparison Test was used for post hoc analysis (ns = non-significant, * *p* < 0.05, ** *p* < 0.01, *** *p* < 0.001, **** *p* < 0.0001; n = 3 independent experiments).

**Figure 6 ijms-23-01371-f006:**
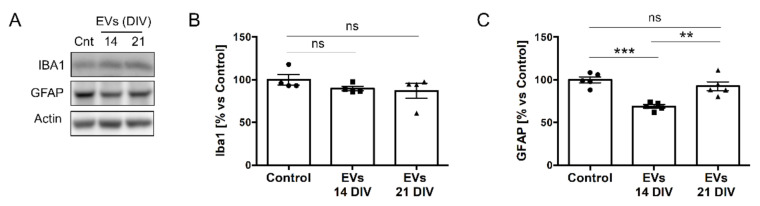
The levels of the astrocytic protein GFAP decrease after treatment with 14 DIV EVs. (**A**) Western blot analysis of microglial marker Iba1 and the astrocytic marker GFAP in 14 DIV neural culture lysates untreated (cnt) or treated for 24 h with EVs isolated from the media of 14 DIV or 21 DIV neural cultures. Actin was used as a loading control. (**B**) Plot comparing the levels of Iba1 quantified from Western blot experiments like the one shown in panel A (n = 4 independent experiments). (**C**) Plot comparing the levels of GFAP, quantified from Western blot experiments like the one shown in panel A (n = 5 independent experiments). All graphs show the mean protein levels relative to the untreated (control) condition ± SEM. Statistical significance was analyzed by one-way ANOVA. Tuckey’s Multiple Comparison Test was used for post hoc analysis (** *p* < 0.01, *** *p* < 0.001, ns = non-significant).

**Figure 7 ijms-23-01371-f007:**
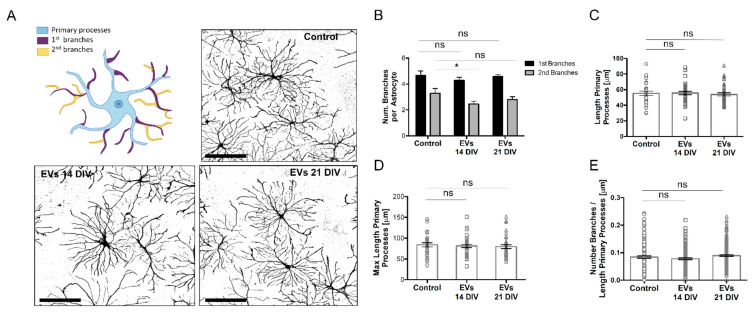
Inflammatory effect of 14 and 21 DIV isolated EVs on 14 DIV neural cultures. (**A**) Representative confocal images of astrocytic processes, detected with an antibody against GFAP, of 14 DIV neural cultures untreated (control) or treated for 24 h with EVs isolated from the media of 14 DIV (EVs 14 DIV) or 21 DIV (EVs 21 DIV) neural cultures. A scheme highlighting the primary processes, primary branches, and secondary branches of astrocytes is shown on the left upper quadrant of this panel. 14 DIV neural cultures untreated (control) or treated for 24 h with EVs 14 DIV or EVs 21 DIV, were analyzed for the number of primary and secondary branches per astrocyte (bar = 100 µm) (**B**), the length of primary astrocytic processes (µm) (**C**), the maximum length of the primary astrocytic processes (µm) (**D**) and the density of the branches on the main processes (number of primary branches normalized to the length of the primary astrocytic processes (num/µm); panel (**E**). All graphs show mean values ± SEM. Statistical significance was analyzed by one-way ANOVA. Tuckey’s Multiple Comparison Test was used for post hoc analysis (* *p* < 0.05, ns = non-significant; n = 3 independent experiments).

**Table 1 ijms-23-01371-t001:** List of antibodies used in Western blot analysis.

Target	Company	Catalogue Number	Host	Dilution
Akt	Cell Signaling Technology	9272S	Rabbit	1:1000
GFAP	Millipore	MAB3402	Mouse	1:1000
p-Akt (S473)	Cell Signaling Technology	4060S	Rabbit	1:1000
PHF Tau	Given by Prof. Jesus Ávila			1:1000
p-p38 (T180/Y182)	Cell Signaling Technology	4511S	Rabbit	1:1000
PSD-95	BD Transduction Laboratories	610495	Mouse	1:500
Secondary goat anti-mouse	DAKO	P0447	Goat	1:2500
Secondary goat anti-rabbit	DAKO	P0448	Goat	1:2500
Synaptophysin	Synaptic Systems	101004	Guinea Pig	1:1000
Tau5	Invitrogen	AHB0042	Mouse	1:500
TrkB	BD Transduction Laboratories	610101	Mouse	1:1000
β-Actin	Abcam	ab8227	Rabbit	1:5000

## Data Availability

The data presented in this study are available on request from the corresponding author.

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
