# Peer review of "Extracellular Vesicles Derived from Young Neural Cultures Attenuate Astrocytic Reactivity In Vitro"

_ijms, 2022, doi:10.3390/ijms23031371_

Round 1

Reviewer 1 Report

In this work, the effect of extracellular vesicles (EVs) derived from mixed neuronal cultures of 14 and 21 DIV on astrocyte morphology and inflammatory status are studied. In such a way, age-depend effects of an “overdose” of EVs on the same culture type are examined. A characterization of protein expression levels in the cultures and of cytokines and LDH in the cell media was performed. They conclude that 14 DIV EVs from these mixed cultures have anti-inflammatory effects.

This study may add knowledge to the field. However, several main points have to be addressed:

  • How do the authors know that EVs are derived from neurons as they state? Indeed, in line 97 it is stated that “….contain mostly neurons but also other cell types”. The cellular composition of the used cultures can easily be determined using markers of each cell type (as in Fig 3).

 At least, to support that EVs are derived from neurons, the effect of EVs derived from purer neuronal cultures should be shown, e.g. inhibiting astrocyte growth by the inhibitor of astrocyte proliferation cytosine arabinofuranoside (AraC). Or the rationale behind the experimental strategy should be explained

If the authors are studying the effect of mixed brain cell derived EVs on astrocytes in these same cultures, why they did not use pure astrocyte cultures. The used experimental design is unable to distinguish between A) whether EVs are derived from neurons B) whether EVs act on neurons and these in turn indirectly affect astrocyte branching through different signaling systems. C) whether mixed EVs act directly on astrocytes (this can be studied in pure astrocyte cultures) D) It is unknown whether the parent cells (cultures of 14 or 21 DIV) show changes without addition of EVs (e.g. comparing GFAP levels in control cultures of 14 and 21 DIV; and which is the relative GFAP content in the derived EVs?). Is the change due to less GFAP transfer by EVs?

  • It has been reported that astrocyte EVs induce a decrease of neuronal branching (dendritic complexity) 1. In this case, EVs were derived from pure astrocyte cultures. The results suggest that reduced branching is a general (not cell-type specific) effect of astrocyte (but not neuron) EVs. Did the authors analyze branching of neurons? These previous findings decrease the novelty of this paper.
  • Characterization of EVs: did the content of classical EV markers change at 14 and 21 DIV? (CD63, CD9, etc). How did the GFAP content change in EVs? (And the content of cytokines, etc, in them?).
  • The consideration of 21 DIV cultures as “aged” needs more information. When comparing 21 DIV with 14 DIV cells, age-related effects can be observed but not necessarily these are “aged” cultures (other authors consider them as mature cultures)

Minor:

Please translate Spanish terms used in the text, e.g. sacarose is sucrose, antibody list: “el que hay es de inmuno” etc

Remove Scheme 1 (now supplementary figure 1) from the main text.

  1. Luarte, A. et al. Astrocyte-Derived Small Extracellular Vesicles Regulate Dendritic Complexity through miR-26a-5p Activity. Cells. 4 (2020).

Reviewer 2 Report

I find that the work is well written and very accurate in the experimental
details. To improve its quality I would suggest to implement discussion with comments on the mechanism by which EVs from older cells have less effect on astrocytes.

I would also suggest removing the description of the extraction method
from the results and leaving it in the methods.

Author Response

We thank the reviewer for the comments on our study. We have followed the reviewer's indications and i) added to the discussion possible causes of the lower effect of EVs from older cultures and ii) changed the site where we describe the EV extraction method, from results to methods.

Round 2

Reviewer 1 Report

The concerns addressed by this reviewer have been adequately addressed. Please revise and add relevant literature in the field (discussion) 

Author Response

We have revised the references of the discussion section and we have added the following references:

  1. Luarte A, Henzi R, Fernández A, Gaete D, Cisternas P, Pizarro M, et al. Astrocyte-Derived Small Extracellular Vesicles Regulate Dendritic Complexity through miR-26a-5p Activity. Cells. 2020 Apr 9(4)930.
  1. Scheller J, Chalaris A, Schmidt-Arras D, Rose-John S. The pro- and anti-inflammatory properties of the cytokine interleukin-6. Biochim Biophys Acta. 2011 May 1813:878–88. 
  1. Marz P, Heese K, Dimitriades-Schmutz B, Rose-John S, Otten U. Role of interleukin-6 and soluble IL-6 receptor in region-specific induction of astrocytic differentiation and neurotrophin expression. 1999 Glia. May 26(3):191–200.
  1. Marz P, Herget T, Lang E, Otten U, Rose-John S. Activation of gp130 by IL-6/soluble IL-6 receptor induces neuronal differentiation. The European Journal of Neuroscience. 1997 Dec 9(12):2765–2773.
  1. Sawada M, Itoh Y, Suzumura A, Marunouchi T. (1993). Expression of cytokine receptors in cultured neuronal and glial cells. Neuroscience Letters. 1993 Oct 160(2):131–134.